# Influence of a Composite Polylysine-Polydopamine-Quaternary Ammonium Salt Coating on Titanium on Its Ostogenic and Antibacterial Performance

**DOI:** 10.3390/molecules28104120

**Published:** 2023-05-16

**Authors:** Lei Xing, Hongyang Song, Jinjian Wei, Xue Wang, Yaozhen Yang, Pengbo Zhe, Mingming Luan, Jing Xu

**Affiliations:** 1Shandong Provincial Key Laboratory of Molecular Engineering, School of Chemistry and Chemical Engineering, Qilu University of Technology (Shandong Academy of Sciences), Jinan 250353, China; xl2713698788@163.com (L.X.); hy2495756023@163.com (H.S.); wxue202111@163.com (X.W.); 15266726773@163.com (Y.Y.); 18991783750@163.com (P.Z.); luanmm@qlu.edu.cn (M.L.); 2College of Chemistry, Chemical Engineering and Materials Science, Shandong Normal University, Jinan 250100, China

**Keywords:** Ti, polylysine, self-assembly technology, EPTAC, DEQAS, MPA-N^+^, cell viability, antibacterial properties

## Abstract

Thin oxide layers form easily on the surfaces of titanium (Ti) components, with thicknesses of <100 nm. These layers have excellent corrosion resistance and good biocompatibility. Ti is susceptible to bacterial development on its surface when used as an implant material, which reduces the biocompatibility between the implant and the bone tissue, resulting in reduced osseointegration. In the present study, Ti specimens were surface-negatively ionized using a hot alkali activation method, after which polylysine and polydopamine layers were deposited on them using a layer-by-layer self-assembly method, then a quaternary ammonium salt (QAS) (EPTAC, DEQAS, MPA-N^+^) was grafted onto the surface of the coating. In all, 17 such composite coatings were prepared. Against *Escherichia coli* and *Staphylococcus aureus*, the bacteriostatic rates of the coated specimens were 97.6 ± 2.0% and 98.4 ± 1.0%, respectively. Thus, this composite coating has the potential to increase the osseointegration and antibacterial performance of implantable Ti devices.

## 1. Introduction

When Ti is used as an implant material, the inert oxide coating on its surface prevents it from having an “active repair function,” thus, rendering the surface vulnerable to bacterial growth. This decreases the biocompatibility between the implant material and the bone tissue, resulting in a decreased rate of osseointegration [1,2,3,4]. Ti has high corrosion resistance in most alkali or salt solutions, although its surface can corrode in an acidic environment. Additionally, Ti has good wettability, and its surface easily adsorbs pollutants [5,6]. Surface modification technology is widely used to address the above problems of rapid breeding of bacteria and surface pollution [7,8]. There are several methods currently available for Ti surface modification, such as self-assembly or layer-by-layer (LBL) technology, anodic oxidation technology, pulse laser method, magnetron sputtering, ion beam assisted deposition, sol-gel, thermal spraying, dip coating, and plasma spraying [9]. LBL technology may be used to produce nanoscale multifunctional composite coatings that optimize the surface properties of Ti [10,11]. A single-layer film or a multi-layer film could be deposited using this technology, which fixes required functional molecules as proteins, peptides, enzymes, drugs, and organic polymers, on the surface [12].

Polylysine (PLL) has strong biocompatibility, and its interaction with cells could be strengthened by using electrostatic principles to promote cell adhesion [13,14,15]. Additionally, some cells can digest and absorb PPL because of its strong penetration into the biofilm [16,17,18]. Therefore, PLL has been widely used in the medical and pharmaceutical fields. According to Kim et al. [19], PLL could promote chondrocyte adhesion and cell functionality. Guo et al. [20] used dopamine (DA) to reduce silver in order to create a silver-loaded composite coating through polyelectrolyte electrostatic self-assembly. Electrostatic interaction can be used to enhance PLL interaction with cells and promote cell adhesion. Sodium alginate (SA) has been extensively used in the field of tissue engineering [21,22,23]. A PLL/SA/PLL(PSP) polyelectrolyte coating was formed on the surface of Ti treated with hot alkali to improve cell compatibility, and the properties of the coating were modified through polymerization, forming polydopamine (PDA) in DA solution. Silver nanoparticles (AgNPs) are formed on the surface by in situ reduction of Ag^+^. Such a coating has an excellent antibacterial effect and good cell compatibility, which may have potential value for dental and orthopedic-related applications. In addition to promoting cell adhesion, PLLs also exhibit a wide range of antibacterial activities [24]. PLL, as a polymer widely used in the field of biology, has many characteristics, such as good antibacterial performance, high safety performance, high thermal stability, and strong water solubility [25,26,27].

Ti and its alloys are widely used in the orthopedic and dental fields due to their excellent mechanical properties and corrosion resistance [28,29]. Since implant-induced infection is generally resistant to the host’s natural defenses as well as to most antibiotics, it is a common and persistent clinical problem, and such infections often necessitate surgical intervention. The design and development of multifunctional surface biomaterials with the goals of eradicating bacterial infection and inhibiting biofilm formation, as well as promoting osseointegration, have been explored in this field in order to effectively address the challenging problem of peri-implant infection. PLLs are viscous molecules that can penetrate biofilms and form strong electrostatic interactions with anion-bearing substances [30]. Based on these properties, PLL-based carrier research has been widely applied in the medical and pharmaceutical fields [31,32,33,34,35]. The polar groups amino and catechol, which are both abundant in DA, enable it to stick to the surface of bio-based materials with ease, forming a PDA coating through deposition, thereby enhancing the hydrophilicity and biocompatibility of the material. When Ti is used for implants, it lacks osseointegration and antibacterial ability, which reduces its biocompatibility with bone tissue and is likely to cause osseointegration failure. Ti surface can easily be corroded by acid and has good surface wettability, which easily adsorbs pollutants. In order to solve the problems of bacteria and pollution on the surface of Ti-based materials, it is necessary to modify the surface. In the present study, the Ti surface was initially negatively ionized using LBL technology, and then the polylysine-poly (dopamine) coating was prepared using layer-by-layer self-assembly technology with PLL and PDA (Figure 1). A composite coating was made of polylysine and polydopamine, onto which a quaternary ammonium salt (QAS) (EPTAC, DEQAS, MPA-N^+^) was grafted. In total, 17 composite coatings were prepared. The antibacterial, cytotoxicity, cell adhesion, cell proliferation, and antibacterial performance of the coated Ti specimens were determined.

## 2. Results and Discussion

### 2.1. Wettability of Surfaces

An optical contact angle measuring instrument was used to measure the Water Contact Angle (WCA) of the Ti surface after various stages in the deposition of the composite coating (Figure 2). The surface of the as-received and uncoated Ti is hydrophobic, with a WCA of 100 ± 2.0°. The surface of Ti activated by hot alkali has super hydrophilic properties porous Ti (pTi), with a WCA of 7 ± 1.2°. This is because the microstructure of the Ti surface is changed (porous) by hot alkali treatment by introducing a large number of hydroxyl groups. After the multilayer PLL and PDA were deposited, the WCA of the surface increased markedly, with the surface WCA of pTi-(PLL-PDA)_1_ and pTi-(PLL-PDA)_2_ being 22 ± 1.1° and 27 ± 1.2°, respectively, as a result of the joint action of PLL and PDA. The WCA of the surface of the coated Ti increased slightly as the number of layers in the composite coating increased, with pTi-(PLL-PDA)_10_ and pTi-(PLL-PDA)_15_ being 33 ± 1.1° and 34 ± 1.0°, respectively, indicating that the coating was stable. The above results show that PLL and PDA fully covered the porous Ti surface and formed a hydrophilic surface.

### 2.2. Surface Roughness

As shown in Figure 3, the uncovered Ti surface roughness (Ra) is 2.31 nm, and Ra is 14.52 nm when the number of assembled PLL and PDA layers is n = 1. The results show that as n increases, the peak on the material surface gradually becomes smooth, and the Ra of the Ti surface exhibits an increasing trend (Figure 4). Ra is 67.87 nm when n = 15 assembled layers because both PLL and PDA are long macromolecular chain structures; they coat and wind on the peaks on the surface of pTi after being assembled on the surface of pTi using LBL technology, and the pores are gradually filled. The above results show that PLL and PDA were successfully assembled on the surface of pTi, and Ra increases with the increase in n. The rough biomolecular coating on the nano level can stimulate osteogenesis around the implant, which increases bone anchoring and biomechanical stability, thereby reducing the bone-bonding period.

### 2.3. Surface Composition

Figure 5 shows the XPS test results for the Ti surface covered with various composite coatings. The Ti substrate is gradually covered by the self-assembly of PLL and PDA, the peak of Ti 2p (458.5 eV) is reduced to zero, and the peaks of N 1s (400.5 eV) and O 1s (532.5 eV) are gradually enhanced [36,37,38], which confirms that PLL and PDA were successfully modified on the Ti surface.

### 2.4. Molar Grafting Rate

As shown in Table 1 (according to the mathematical formula in 3.8), (2,3-Epoxypropyl) trimethylammonium chloride (EPTAC) has a higher grafting rate than the other two quaternary ammonium salts because its molecular chain is shorter, allowing more molecules to be grafted on the surface of the self-assembled layer. Additionally, the molar grafting rate increases gradually after grafting the quaternary ammonium salt with an increase in the number of self-assembled layers. It was shown that the catechol group in PDA is easily oxidized to a quinone structure under alkaline conditions, and the amino group can be grafted with an epoxy group or carboxyl group.

### 2.5. Cell Compatibility

Cytotoxicity was determined in terms of optical density (OD). For a given specimen, the higher the OD the more cells have proliferated and differentiated on it, and the higher the biological activity of the cells. Figure 6 shows that the cytotoxicity performance is: pTi-(PLL-PDA)_15_-EPTAC > pTi-(PLL-PDA)_15_-DEQAS > pTi-(PLL-PDA)_15_-(MPA-N^+^)/DEQAS > pTi-(PLL-PDA)_15_-(MPA-N^+^). When the cells were cultured for 24 h, compared with the blank specimens, the existence of quaternary ammonium salt as a functionally modified surface had no effect on cell viability and growth. Among the coated specimens, the ones on which MPA-N^+^ grafted had been onto self-assembled multilayers showed the highest cell vitality, exhibiting excellent cell adhesion and proliferation. This shows that the polylysine-polydopamine-quaternary ammonium salt composite coating promotes cell proliferation. The above results show that the composite coating is non-toxic to cells and favorable to the biocompatibility of cells.

### 2.6. Antibacterial Performance

Antibacterial performance was determined using the plate colony counting method. Plain or uncoated Ti (pTi) attracted a large number of bacteria after 24 h of culture in *E. coli* and *S. aureus* (Figure 7) and, as such, had very poor antibacterial properties. In contrast, coated Ti specimens in which the coating consisted of a grafted quaternary ammonium salt had excellent antibacterial properties (Figure 7). The index of antibacterial performance is expressed as the kill ratio (defined as % of bacteria that remained on the specimen surface 24 h after cultivation with the test bacterium) (Figure 8). The kill ratios against *E. coli* and *S. aureus* after coating that included grafting with DEQAS were 70.8 ± 3.0% and 92.9 ± 2.0%, respectively, whereas the kill ratios of specimens on which the coating included grafted MPA-N^+^ to *E. coli* and *S. aureus* were 93.1 ± 2.0% and 94.7 ± 2.0%, respectively. The kill ratios of specimens coated with DEQAS and MPA-N^+^ to *E. coli* and *S. aureus* were 97.6 ± 2.0% and 98.4 ± 1.0%, respectively. The kill ratio of specimens whose coating comprised a mixture of grafted DEQAS and MPA-N^+^ was higher than that of those whose coating comprised a single grafted salt of quaternary ammonium. This is because the raw material for MPA-N^+^ synthesis had anti-inflammatory properties, and the antibacterial property of two quaternary ammonium salts was higher than that of one quaternary ammonium salt.

Based on the composite coating as an implant material, we only used a single HUVECs cell to detect biocompatibility during the study and did not select multiple cells. The selected antibacterial species are not perfect, and the flora affecting peri-implantitis should be mainly anaerobic bacteria. For example, corresponding antibacterial experiments should be performed on *Porphyromonas gingivalis*, *Streptococcus*, etc. Although there are few biological experiments tested, the above data show that the polylysine-polydopamine-quaternary ammonium salt composite coating has good biocompatibility and excellent antibacterial properties.

## 3. Materials and Methods

### 3.1. Materials

Dopamine (AR), (2,3-Epoxypropyl) trimethylammonium chloride (EPTAC, AR), 2-(*N*-morpholinyl) ethanesulfonic acid buffer solution (MES, AR), *N*-hydroxysuccinimide (NHS, AR), *N*,*N*,*N*′,*N*′-Tetramethylethylenediamine (TMEDM, AR), Epichlorohydrin (EC), 1-ethyl-(3-dimethylaminopropyl) carbodiimide (EDC, AR), Abietic acid (AR) were provided by McLin Biochemical Technology Co., Ltd. (Shanghai, China), l-lysine (AR) was provided by Bailingwei Technology Co., Ltd., sodium carbonate (AR), sodium bicarbonate (AR) and Potassium sulfate (AR) were provided by Aladdin Biochemical Technology Co., Ltd. (Shanghai, China), and anhydrous ethanol was provided by Tianjin Fuyu Fine Chemical Co., Ltd. (Tianjin, China), Ti (specification: 1 cm × 1 cm, thickness 1 mm) was provided by Guangdong Als Metal Technology Co., Ltd. (Guangdong, China).

### 3.2. Preparation of Collagen Peptide Solution

Synthesis of DEQAS [39]: The 250.0 mL three-port flask was taken, and distilled water (22.0 mL), K_2_SO_4_ (0.2 g), methanol (17.0 mL), and epichlorohydrin (EC, 9.5 g) were added to it in turn. Then the three-port flask was heated to 50 ± 1 °C and stirred at a constant temperature for 0.5 h. During this time, tetramethylethylenediamine (TMEDM) was added to the flask at a rate of 12 drops/min. The reaction was stopped after stirring for 1.5 h. The solution was poured into a 250 mL round-bottom flask and distilled under reduced pressure to obtain a yellowish liquid (DEQAS) (Figure 9).

Synthesis of MPA-N^+^: The rosin acid-derived quaternary ammonium maleic acid cation (MPA-N^+^) was prepared using a method detailed in the literature [40], and with the reaction route shown in Figure 10. The Abietic acid (100.0 g, 0.28 mol) was heated to 180 °C and refluxed for 3 h under a nitrogen atmosphere, and the heated rosin acid was cooled to 120 °C, after which maleic anhydride (27.5 g, 0.28 mol) and acetic acid (400.0 mL) were added to the above reaction system. The reaction was refluxed at 120 °C for 12 h. Then the reaction was cooled to room temperature and left for another 2 h. The crude maleic pine acid was recrystallized twice in acetic acid to obtain pure maleic pine acid (MPA, 91.0 g, 97% purity, 79% yield).

Maleic acid (MPA, 10.0 g, 0.025 mol) was dissolved in ethanol (250.0 mL), then *N*,*N*-dimethylethylenediamine (2.8 mL, 0.025 mol) was added and refluxed at 85 °C for 5 h. The solution was then cooled to room temperature, crystallized, filtered, and dried to obtain the product MPA-N (9.5 g, purity: 98%, yield: 79%).

MPA-N (1.0 g, 0.0021 mol) and bromoethane (3.1 mL, 0.043 mol) were dissolved in redistilled tetrahydrofuran (THF, 30.0 mL) and reacted at 40 °C for 48 h. Then the product MPA-N^+^ (0.94 g, purity: 90%, yield: 70%) was crystallized, filtered, and dried.

Figure 11 shows the 1H NMR spectrum derived from the structural characterization of MPA, MPA-N, and MPA-N+. ^1^H NMR (DMSO, 400 MHz), δ (TMS, ppm), 12.08 (s, 1H), 5.52 (s, 1H), 3.23 (s, 1H), 2.91 (s, 1H), 2.33 (s, 1H). Figure 11b ^1^H NMR (CDCl3, 400 MHz), δ (TMS, ppm), 12.16 (s, 1H), 5.42 (s, 1H), 3.71 (t, 2H), 3.41 (s, 1H) and 3.19 (s, 1H). Figure 11c ^1^H NMR (DMSO, 400 MHz), δ (TMS, ppm), 12.16 (s, 1H), 5.38 (s, 1H), 3.63 (t, 2H), 3.40 (t, 2H) and 3.17 (s, 2H). 1.68 (m, 11H), 1.45 (s, 4H), 1.23 (t, 3H), 1.04 (s, 3H), 0.90 (t, 6H), 0.53 (s, 3H). The attribution of each peak was determined using information such as various chemical shifts of protons and different chemical environments.

### 3.3. Preparation of Polylysine-Polydopamine-Quaternary Ammonium Salt Composite Coating

Preparation of polylysine-polydopamine coating: High-purity Ti sheets were polished, cleaned, and dried in a constant temperature drying oven at 60 °C for 12 h, and the sheets were designated uncoated or pure Ti (pTi). The Ti sheets were then put into 5 mol/L NaOH solution, activated at 60 °C for 5 h, washed with distilled water to neutral, dried with high purity nitrogen, and dried in a constant temperature drying oven at 60 °C for 12 h. Next, we accurately weighed 250 mg of PLL in a 25 mL small beaker, added an appropriate amount of distilled water to dissolve it, stirred it with a glass rod to accelerate the dissolution, and then transferred it to a 50 mL volumetric flask. The Tris-HCl buffer with a pH of 8.5 ± 0.1 and a concentration of 0.01 mol/L was prepared, and then DA was added to prepare a 2 mg/mL DA solution. The pTi sheet was immersed in a PLL solution with a concentration of 5 mg/mL after undergoing a reaction over a period of 24 h in a dark environment to allow PLL to form a uniform coating. Then, the sheet was immersed in a PDA solution with a concentration of 2 mg/mL after undergoing a reaction over a period of 24 h in a dark environment to allow PDA to form a uniform coating. After assembly, the pTi sheets were washed five times with distilled water, dried with high-purity nitrogen, and stored in nitrogen. The above process was repeated; the last layer was PDA, and the sheet was designated pTi-(PLL-PDA)_n_.

Preparation of polylysine-polydopamine-EPTAC composite coating: Firstly, a Na_2_CO_3_/NaHCO_3_ buffer solution with a pH of 9.6 was prepared, and 5 mL of buffer solution was added to the reaction bottle. Ten drops of EPTAC were added to a 1 mL syringe, and the reaction bottle was ultrasonically treated for 10 min to make EPTAC completely dissolve in the buffer solution. The sample pTi-(PLL-PDA)_n_ was placed in a reaction bottle at a constant temperature of 50 °C for 12 h and then uniformly pulled up and down in distilled water 10 times to remove weakly bound or unbound quaternary ammonium salts. The samples were dried with high-purity nitrogen and stored in nitrogen. The obtained coating was labeled as pTi-(PLL-PDA)_n_-EPTAC.

Preparation of polylysine-polydopamine-DEQAS composite coating: Na_2_CO_3_/NaHCO_3_ buffer solution (pH = 9.6) was prepared, 5 mL buffer solution were poured into the reaction bottle, 10 drops of DEQAS were added to a 1 mL syringe, and the reaction bottle was ultrasonically treated for 10 min to make DEQAS completely dissolve in the buffer solution. The pTi-(PLL-PDA)_n_ sheet was placed in a reaction bottle at a constant temperature of 50 °C for 12 h and then uniformly pulled up and down in distilled water 10 times to remove weakly bound or unbound quaternary ammonium salts. The sheets were dried with high-purity nitrogen and stored in nitrogen. The obtained coating was labeled as pTi-(PLL-PDA)_n_-DEQAS.

Preparation of polylysine-polydopamine-MPA-N^+^ composite coating: 5 mL 2-(*N*-morpholinyl) ethanesulfonic acid (MES) buffer solution (0.1 mol/L, pH = 5.5), EDC(1-3-(dimethylaminopropyl)-3-ethylcarbodiimide hydrochloride), NHS (*N*-hydroxythiosuccinimide) and MPA-N^+^ (n (MPA-N^+^):n (EDC) = 1:427.35) were poured into the a reaction flask. Then (MPA-N^+^):n (NHS) = 1:854.70), the reaction bottle was ultrasonically treated for 10 min so that MPA-N^+^could be completely dissolved in the buffer (the concentration of MPA-N^+^ was 0.0144 mol/L (8.4 mg/mL)); then, the prepared self-assembled composite coating was placed in the above reaction bottle, and the reaction was carried out at 50 °C for 12 h. After that, it was pulled up and down at a constant speed in distilled water 10 times to remove the weakly bound or unbound quaternary ammonium salt, dried with high-purity nitrogen, and stored in nitrogen. The obtained coating was labeled as pTi-(PLL-PDA)_n_-MPA-N^+^.

Preparation of polylysine-polydopamine-DEQAS/MPA-N^+^ composite coating: The pH = 9.6 Na_2_CO_3_/NaHCO_3_ buffer solution was prepared, and 5 mL buffer solution and DEQAS (21.3 mg) were added to the reaction bottle. The above reaction bottle was placed in an ultrasonic cleaner for 10 min so that DEQAS could be completely dissolved in the buffer solution (the concentration of DEQAS was 0.0142 mol/L). Then, the prepared self-assembled composite coating was placed in the above reaction bottle, and the reaction was carried out at a constant temperature of 50 °C for 12 h. After that, it was uniformly pulled up and down 10 times in distilled water to remove the weakly bound or unbound quaternary ammonium salt. The coating was dried with high-purity nitrogen and stored in nitrogen. The obtained coating was labeled as pTi-(PLL-PDA)_n_-DEQAS. Then another reaction bottle was taken, and 5 mL buffer solution was added, EDC(1-(3-dimethylaminopropyl)-3-ethylcarbodiimide hydrochloride), NHS (*N*-hydroxythiosucc-inimide) and MPA-N^+^ (20.30 mg) (n (MPA-N^+^):n (EDC) = 1:427.35); the n (MPA-N^+^):n (NHS) = 1:854.70), the above reaction bottle was placed in an ultrasonic cleaner for 10 min, so that MPA-N^+^ could be completely dissolved in the buffer solution (the concentration of MPA-N^+^ was 0.00173 mol/L); then, the prepared self-assembled coating pTi-(PLL-PDA)_n_-DEQAS was placed in the above reaction flask, and the reaction was carried out at a constant temperature of 50 °C for 12 h. The coating was uniformly pulled up and down 10 times in distilled water, and the weakly bound or unbound quaternary ammonium salt was removed, dried with high-purity nitrogen, and stored in nitrogen. The obtained coating was labeled as pTi-(PLL-PDA)_n_-DEQAS/(MPA-N^+^).

### 3.4. Determination of UV-Visible Spectra

Eight 30 mL sample bottles were used to configure an equal volume and equal concentration of DA solution (concentration of 2 mg/mL), which were irradiated under an ultraviolet lamp for 0 min, 15 min, 30 min, 60 min, 120 min, 180 min, 240 min, and 300 min, respectively. Then, 3 mL were taken out with a pipette and put into a cuvette, and the ultraviolet test was carried out by using a UV spectrophotometer (UV-1800PC).

Visible UV Characterization of DA Solution: PDA may be applied to any substrate, and when exposed to ultraviolet radiation, the PDA layer can produce free radicals, thus initiating the polymerization process. The substrate was immersed in DA solution (2 mg/mL, 10 mM Tris HCl, pH = 8.5), and PDA deposition was triggered by ultraviolet irradiation (36 W lamp), allowing spontaneous deposition of PDA film, which was very important for a well-controlled deposition process. This process was different from the slow process of early dynamics, and PDA was triggered to produce free radicals for the polymerization of various monomers under further sunlight irradiation. Figure 12 shows the color change of the DA solution at different reaction times under ultraviolet and dark conditions. After 2 h, the UV-irradiated solution became darker, but the non-irradiated solution showed a gradual color change. Figure 13 shows the absorption spectrum of DA solution under ultraviolet irradiation, indicating that the absorption peak increased around 410 nm and increased progressively with the polymerization time of PDA.

In this paper, different coatings were tested differently, as shown in Table 2. The specific experimental steps are as follows:

### 3.5. Determination of WCA

The water contact angle (WCA (θ)) value can be used to directly determine the hydrophilicity of the surface. When θ is less than 90°, the surface of the material can be wetted by liquid (Figure 14a), which is hydrophilic. The smaller the θ angle, the better the wettability of the surface. When θ is greater than 90°, the surface of the material cannot be wetted by liquid (Figure 14b), which is hydrophobic. The larger the θ angle, the higher the hydrophobicity of the surface [41,42,43,44,45].

At room temperature, WCAs of the above prepared Ti specimens were measured using an optical contact angle measuring instrument (DSA-100, Kruss, Germany). The distilled water of the automatic distribution controller was dripped onto the sample to be tested (~5 μL), and five different positions were dripped on each sample. After 20 s (the contact angle decreases with time, and the optimum measurement time is 20 s), the Laplace–Young fitting algorithm was used to determine the mean value of WCA, click on the photo, and observe and record the WCA of the image.

### 3.6. AFM Characterization

The surface morphology of the prepared samples was characterized by AFM (Multimode8, Bruker, Germany). The Ra of the coating surface was measured by AFM, and the specimens were characterized using the Peak Force mode. The scanning range of AFM was set to 1 μm, and the scanning speed was set to 0.977 Hz. Five different regions were selected for a sample to preserve the data with uniform distribution of surface morphology changes. A software package (NanoScope Analysis) was used to process the acquired data.

### 3.7. X-ray Photoelectron Spectroscopy (XPS)

The prepared specimens were examined using X-ray photoelectron spectroscopy (ESCALABXi+, Thermo, Waltham, MA, USA, Microfocusing monochromatic (Al Kα) X-ray source (the best sensitivity of monochromatic light source: 1600 kcps (Al Kα 0.60 eV Ag 3d5/2 peak)) at an angle of incidence of 30° (measured from the surface) and an emission angle normal to the surface. The power used was 150 W, and the maximum resolution depth was 10 nm. Survey spectra (Binding Energy (BE) in the range of 0–5000 eV) were used for element identification and quantification. The middle position of the specimen was scanned using a high magnification mode. The specimens pTi, pTi-(PLL-PDA)_1_, pTi-(PLL-PDA)_5_, pTi-(PLL-PDA)_10_, and pTi-(PLL-PDA)_15_ were analyzed by composition (0–5000 eV) and high-resolution energy spectra (N 1s, O 1s). All reported spectra are averages of five scans taken at a resolution of 0.1 eV and referenced to the C 1s peak of hydrocarbons at 284.8 eV. Data acquisition and processing were performed using Thermo Advantage software. The XPS spectra were fitted with Voigt profiles obtained by convolving Lorentzian and Gaussian functions. The analyzer transmission function, Scofield sensitivity factors, and effective attenuation lengths (EALs) for photoelectrons were applied for quantification. EALs were calculated using the standard TPP-2M formalism.

### 3.8. Molar Grafting Rate

In order to simulate a Ti sheet, pure Ti powder was pressed into a 75 mm diameter disk. A thin layer of Ti was deposited on a quartz disk (diameter: 25 mm) by reactive magnetron sputtering using a radio-frequency magnetron sputtering system (CFS-4ES-231). The magnetron sputtering chamber was evacuated to a certain pressure and kept at 6.7 × 10^−1^ Pa with argon, resulting in a Ti layer. The Ti layers were cleaned with sodium dodecyl sulfate and UV-ozone cleaner before the QCM measurements. In QCM, Δ*F* depends on the adsorbed mass following Sauerbrey’s equation:$$\Delta F=-\frac{2F_{0}^{2}}{\sqrt{\rho_{q}\mu_{q}}}\frac{\Delta m}{A}$$
where F_0_ is the fundamental frequency of the crystal (27 × 10^6^ Hz), A is the electrode area (0.049 cm^2^), *ρ_q_* is the quartz density (2.65 g/cm^3^), and *μ_q_* is the shear modulus of quartz (2.95 × 10^11^ dyn/cm).

The measurements were taken in triplicate. The molar grafting rate of the quaternary ammonium salt was calculated by the following formula [46].
(1)$$\mathrm{molar}\;\mathrm{grafting}\;\mathrm{rate}=\frac{W_{D}-W_{0}}{{M_{W}\times W}_{0}}\times 100\%$$
where W_D_ is the mass after grafting quaternary ammonium salt, W_0_ is the mass before grafting quaternary ammonium salt, and M_W_ is the molecular mass of the quaternary ammonium salt. The standard deviation is ±0.001.

### 3.9. Cell Compatibility Determination

Cytotoxicity was obtained using the 3-(4,5-Dimethyl-2-Thiazolyl)-2,5-Diphenyl Tetrazolium Bromide (MTT) assay. Pure Ti, pTi-(PLL-PDA)_15_-DEQAS, pTi-(PLL-PDA)_15_-MPA-N^+^, pTi-(PLL-PDA)_15_-DEQAS/MPA-N^+^, and pTi-(PLL-PDA)_15_-EPTAC were placed in the pores of the cell culture plate. HUVECs (2 × 10^5^ cells/well) were seeded in the wells of the cell culture plate. The specimens were placed in RPMI 1640 medium with a temperature of 37 °C, a CO_2_ concentration of 5%, and a fetal bovine serum (FBS) content of 10% for 24 h. The cells were washed twice with the essential medium Eagle (MEM) (serum-free) after 24 h, and then 15 mL of MTT solution was added to the wells of the cell culture plate. The cells were cultured at a temperature of 37 °C and a CO_2_ concentration of 5% for 1 h. The samples were clipped out with tweezers and placed in the wells of the new cell culture plate. A pipette was used to add 200 μL of DMSO to each well. The well plate was shaken manually in an even manner. After waiting for 10 min, the optical density (OD) value of the mixed product was measured by a microplate reader at a wavelength of 490 nm. Finally, the cell viability was calculated by the following formula.
Viability = (Specimen abs/Control abs) × 100(2)

The OD value of various liquids was measured by a microplate reader because MTT can be reduced to blue crystalline formazan by succinate dehydrogenase in the mitochondria of cells, which can be dissolved by DMSO. Therefore, the larger the OD value of the mixed solution measured by the microplate reader, the more the number of cells and the more the proliferation. All measurements were taken in triplicate.

### 3.10. Antibacterial Performance Determination

When *E. coli* and *S. aureus* grew to the mid-log phase, the bacterial suspension was diluted to 10^6^ CFU/mL. Pure Ti, pTi-(PLL-PDA)_15_-DEQAS, pTi-(PLL-PDA)_15_-MPA-N^+^, and pTi-(PLL-PDA)_15_-DEQAS/MPA-N^+^ were cultured in 5 mL bacterial suspension at 37 °C for 24 h. After incubation, the specimens were washed twice with PBS. The specimens were soaked in 5 mL of PBS for 5 min, and then 3 mL of the soaked bacterial solution was taken by a pipette and centrifuged in a centrifuge tube for 1 min. Then the lower bacterial solution was taken out and evenly spread in a Mueller–Hinton agar plate medium for 12 h. The kill ratio (or antibacterial index) was calculated as follows:AR_kill ratio_ (%) = (CFU_control_) − (CFU_experiment_)/(CFU_control_) × 100%(3)

## 4. Conclusions

The following are the main conclusions drawn from the results obtained.

1. In the work, the Ti surface was made negative using the hot alkali activation method, and then, a poly-lysine and poly-dopamine multilayer was deposited on it using the layer-by-layer self-assembly method.

2. The water contact angle (WCA) on the Ti surface decreased from 100 ± 2.0° to 7.0 ± 1.2° after hot alkali activation, indicating a marked increase in its hydrophilicity. After coating the surface with a poly-lysine-poly-dopamine multilayer, WCA increased, with an increase in the number of layers (n). With n = 15, WCA was stable, being 34.0 ± 1.0°.

3. The surface roughness (Ra) of the coated specimens increased from 14.52 nm to 67.87 nm when n was increased from 1 to 15. The XPS results indicated that the -COOH exposure increased from 4.90% to 17.34%, and that of -NH_2_ increased from 6.48% to 23.48%. The coating that included a grafted quaternary ammonium salt has excellent biocompatibility, with bacteriostatic rates of 97.6 ± 2.0% and 98.4 ± 1.0% against *Escherichia coli* and *Staphylococcus aureus*, respectively.

4. Implantable Ti components/devices on which coatings produced are deposited have the potential to increase both their osseointegration and antibacterial performance.

5. This paper provides ideas for the application of macromolecules in the field of new materials through self-assembly technology. However, due to factors such as test conditions and time, there are some defects in the overall experimental research ideas, experimental process, and test characterization. Based on the research content of this paper, follow-up research work can be carried out in the following ways: In the present study, the self-assembled design of the sample was not applied to the entity animal to detect its planting effect; the molecules used are natural biological macromolecules, which do not involve the study of the degradation of the sample coating and cannot predict whether the degradation of the coating surface will cause adverse reactions.

## Figures and Tables

**Figure 1 molecules-28-04120-f001:**
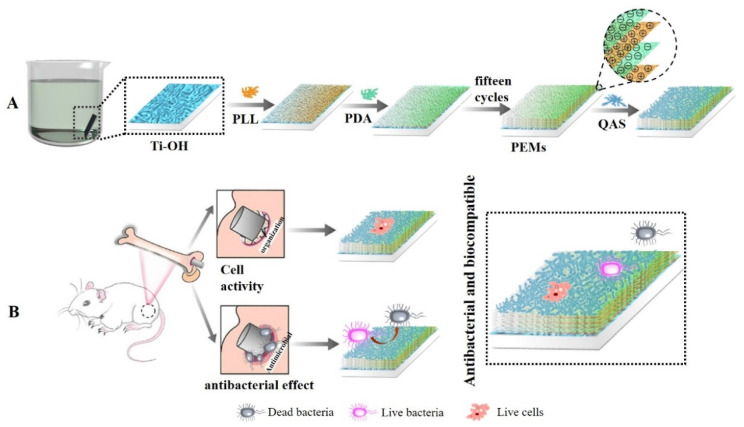
Mechanism diagram of polylysine-polydopamine coating prepared by layer-by-layer self-assembly technique. (**A**) Polylysine-polydopamine coating prepared; (**B**) Coated Ti specimens as implants.

**Figure 2 molecules-28-04120-f002:**
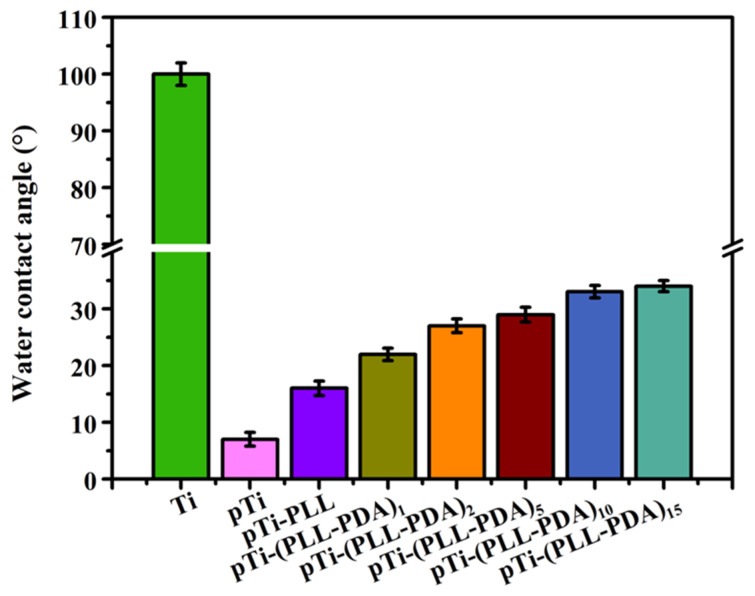
Summary of the water contact angle results. SD 0.31, n = 5.

**Figure 3 molecules-28-04120-f003:**
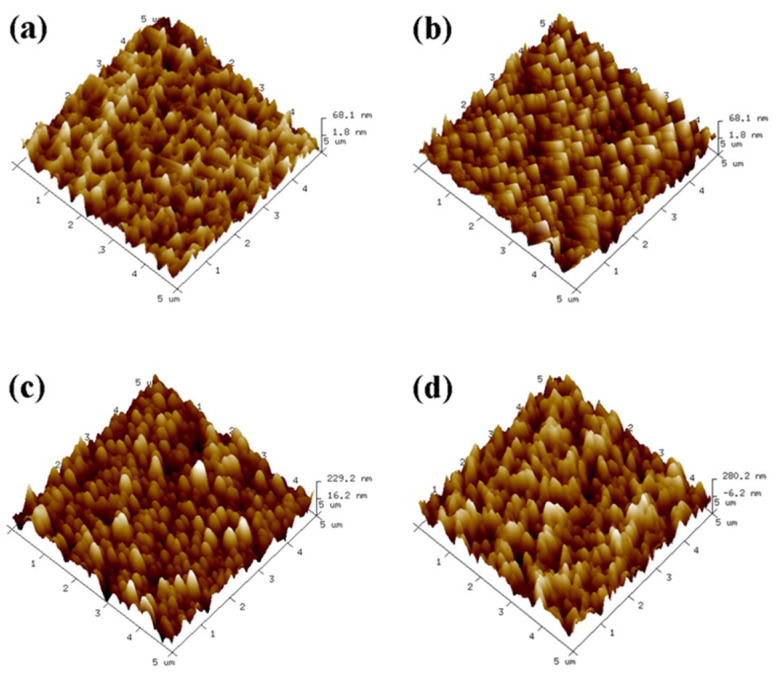
Three−dimensional topography of the Ti surface after self−assembly of polylysine−polydopamine coatings with a different number of assembled layers ((**a**): pTi−(PLL−PDA)_1_; (**b**): pTi−(PLL−PDA)_5_; (**c**): pTi−(PLL−PDA)_10_; (**d**): pTi−(PLL−PDA)_15_).

**Figure 4 molecules-28-04120-f004:**
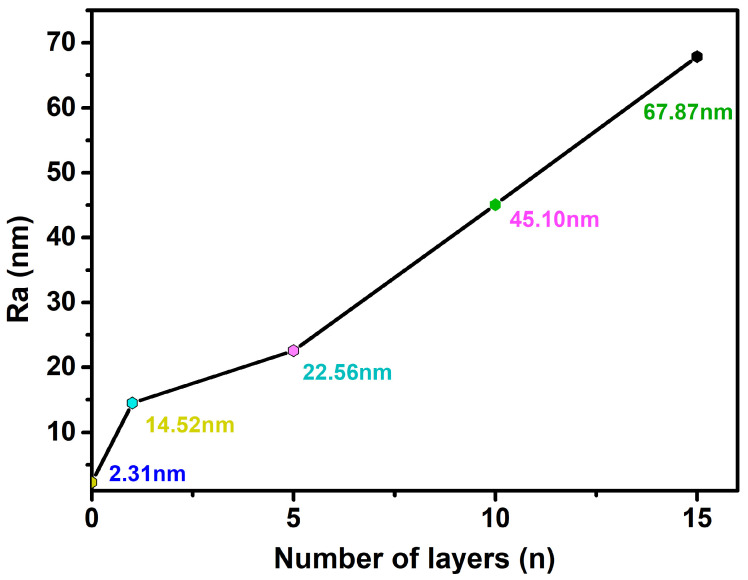
Ra of Ti surface after coating with various layers (n) of self-assembled PLL and PDA.

**Figure 5 molecules-28-04120-f005:**
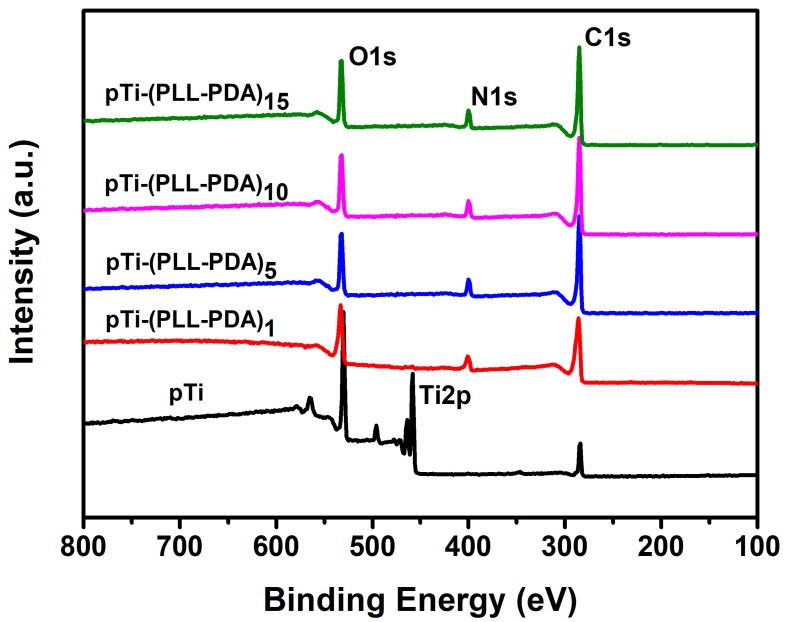
XPS spectra of pTi before and after different coatings.

**Figure 6 molecules-28-04120-f006:**
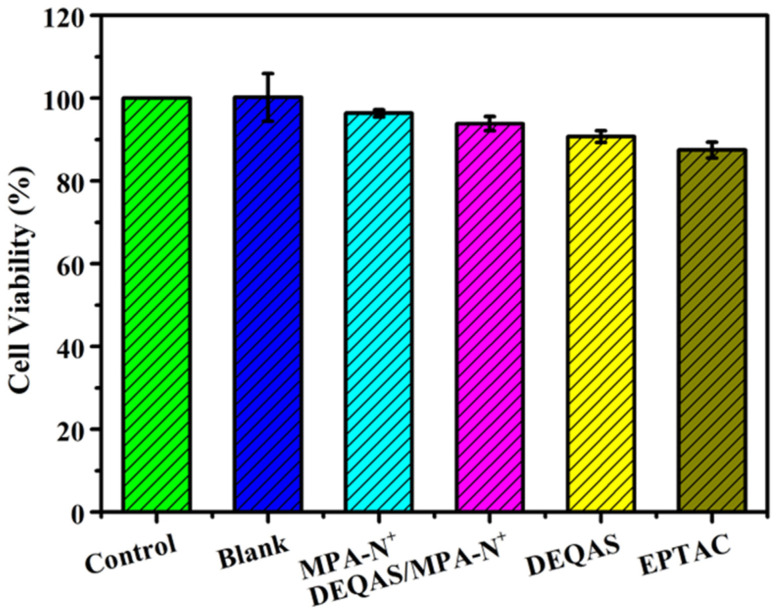
Summary of the cell viability results. SD 1.99, n = 3.

**Figure 7 molecules-28-04120-f007:**
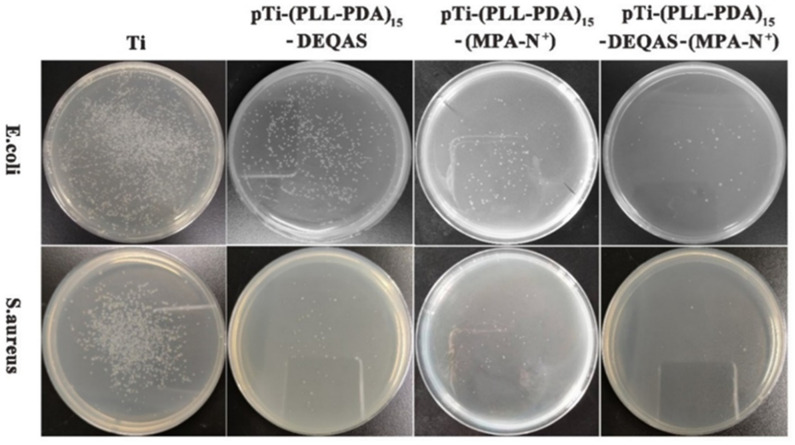
Colony images of *E. coli* and *S. aureus* on the surfaces after 24 h of cultivation of the bacterium.

**Figure 8 molecules-28-04120-f008:**
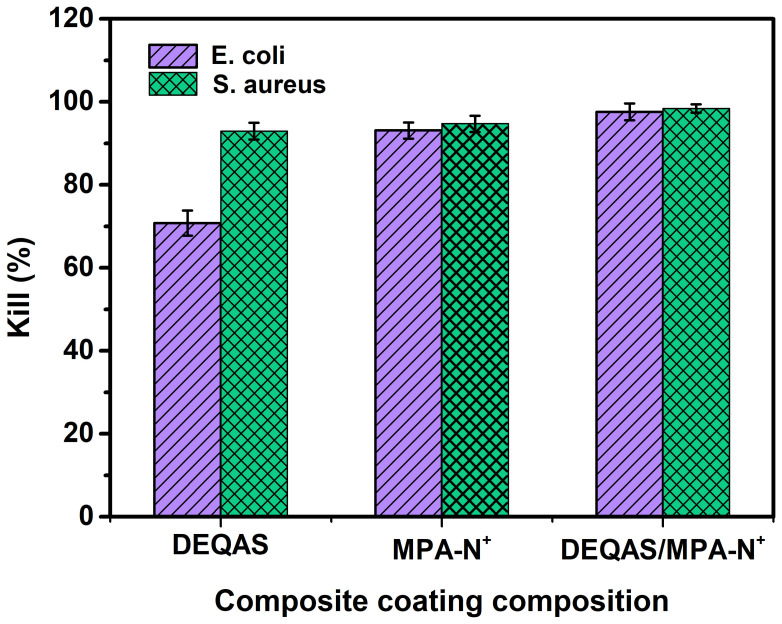
Bacteriostatic rates of coated Ti specimens (layers = 15) against *E. coli* and *S. aureus* after 24 h cultivation of the bacterium. SD 0.58, n = 3.

**Figure 9 molecules-28-04120-f009:**
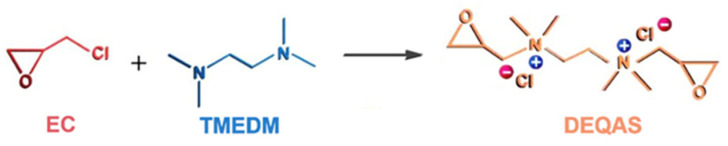
Roadmap for the synthetic reaction for DEQAS.

**Figure 10 molecules-28-04120-f010:**
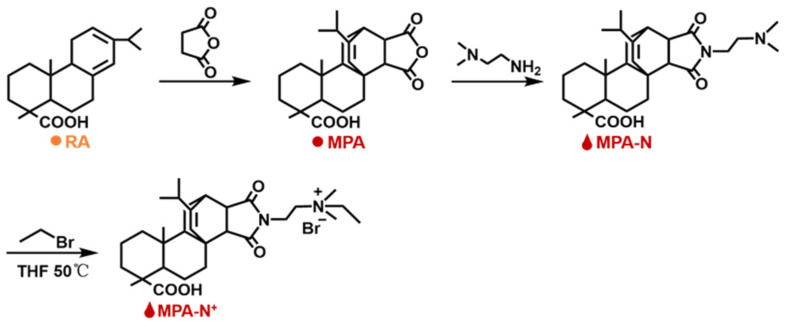
Roadmap for the synthesis reaction for MPA-N^+^. ^1^H NMR Characterization of DEQAS and MPA-N^+^: Refer to our previous literature for DEQAS synthesis.

**Figure 11 molecules-28-04120-f011:**
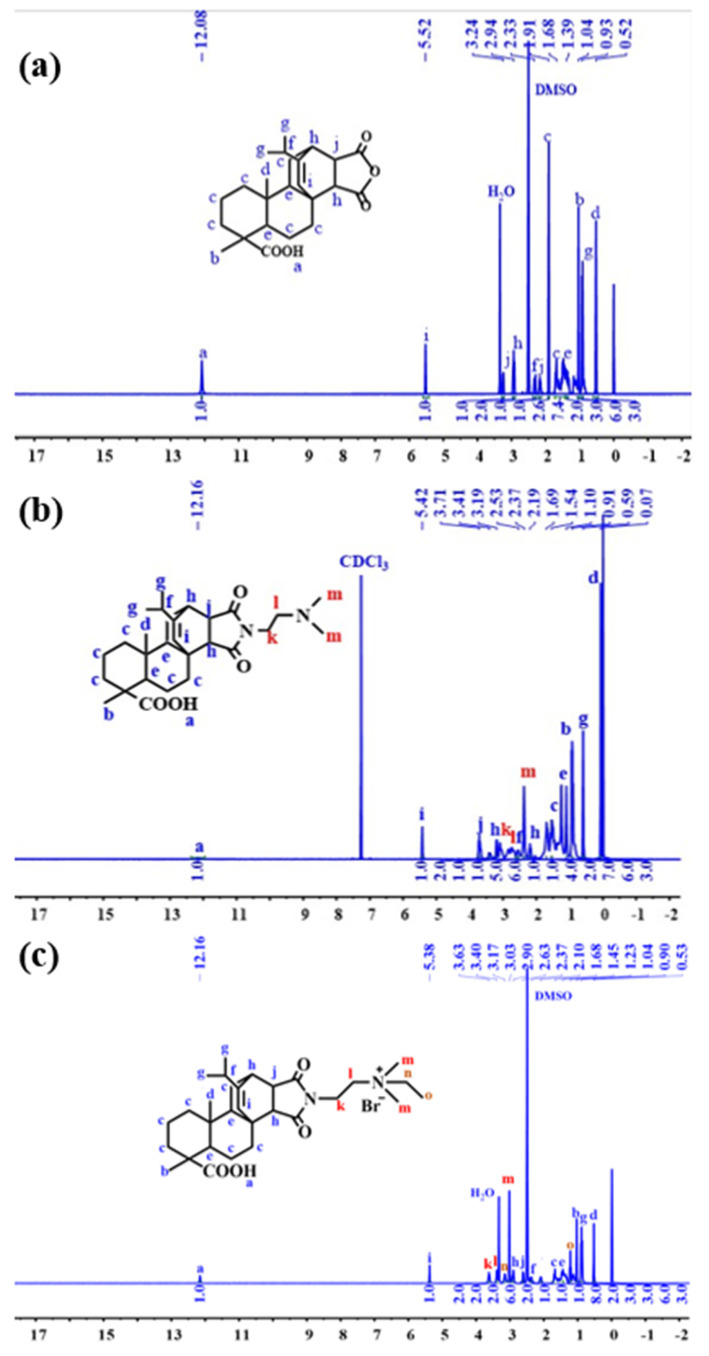
^1^H NMR Spectra of MPA, MPA-N, and MPA-N^+^ ((**a**), MPA, (**b**), MPA-N, (**c**), MPA-N^+^).

**Figure 12 molecules-28-04120-f012:**
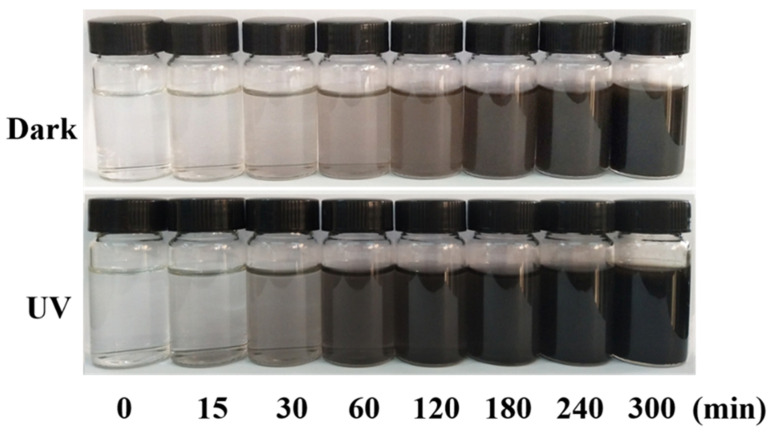
Color changes of DA solution under dark (**top**) and ultraviolet (**bottom**) conditions.

**Figure 13 molecules-28-04120-f013:**
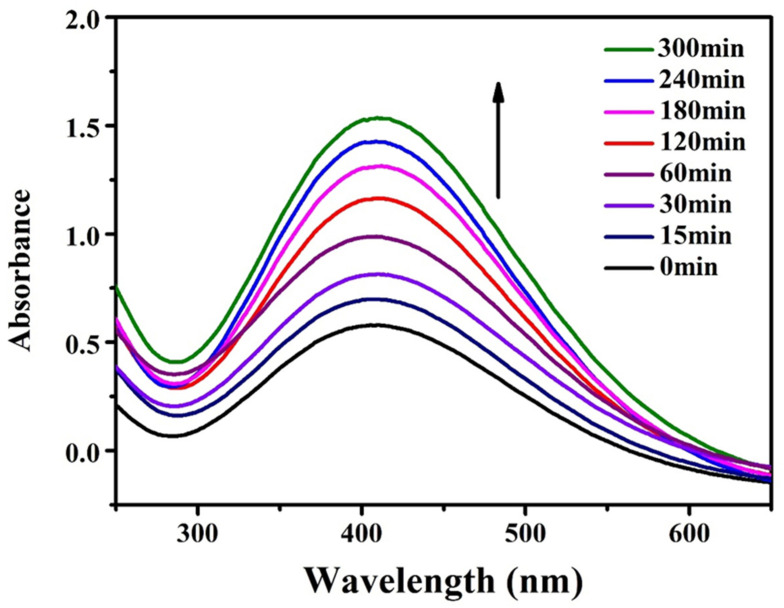
Variation of absorption spectrum of DA solution with reaction time under ultraviolet irradiation.

**Figure 14 molecules-28-04120-f014:**
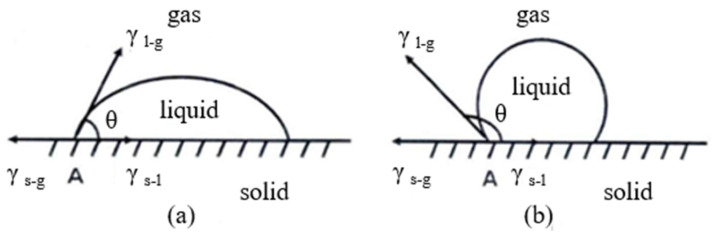
Droplet shape and WCA (l: liquid; g: gas; s: solid). (**a**) θ < 90° is hydrophilic; (**b**) θ > 90° is hydrophobic.

**Table 1 molecules-28-04120-t001:** Molar Grafting Rate of Different Quaternary Ammonium Salts under Different Conditions (%).

Sample	QAS				
	EPTAC	DEQAS	MPA-N^+^	DEQAS/MPA-N^+^	
				DEQAS	MPA-N^+^
pTi-(PLL-PDA)_1_	4.074	3.732	4.936	4.839 (±0.001)	1.862 (±0.001)
pTi-(PLL-PDA)_5_	5.752	4.573	4.830	5.109 (±0.001)	1.959 (±0.001)
pTi-(PLL-PDA)_10_	7.103	6.291	5.138	7.480 (±0.001)	3.162 (±0.001)
pTi-(PLL-PDA)_15_	7.940	6.493	5.392	7.529 (±0.001)	3.257 (±0.001)

**Table 2 molecules-28-04120-t002:** Different properties tested by different types of layers.

Study Group	Properties Determined			
	WCA	Grafting Rate	Cell Viability	Kill%
As-received, uncoated Ti	+		+	+
Porous Ti (pTi)	+			
pTi-PLL	+			
pTi-(PLL-PDA)_1_	+			
pTi-(PLL-PDA)_2_	+			
pTi-(PLL-PDA)_5_	+			
pTi-(PLL-PDA)_10_	+			
pTi-(PLL-PDA)_15_	+			
pTi-(PLL-PDA)_1_-EPTAC		+		
pTi-(PLL-PDA)_1_-DEQAS		+		
pTi-(PLL-PDA)_1_-(MPA-N^+^)		+		
pTi-(PLL-PDA)_1_-(MPA-N^+^)/DEQAS		+		
pTi-(PLL-PDA)_5_-EPTAC		+		
pTi-(PLL-PDA)_5_-DEQAS		+		
pTi-(PLL-PDA)_5_-(MPA-N^+^)		+		
pTi-(PLL-PDA)_5_-(MPA-N^+^)/DEQAS		+		
pTi-(PLL-PDA)_10_-EPTAC		+		
pTi-(PLL-PDA)_10_-DEQAS		+		
pTi-(PLL-PDA)_10_-(MPA-N^+^)		+		
pTi-(PLL-PDA)_10_-(MPA-N^+^)/DEQAS		+		
pTi-(PLL-PDA)_15_-EPTAC		+	+	
pTi-(PLL-PDA)_15_-DEQAS		+	+	+
pTi-(PLL-PDA)_15_-(MPA-N^+^)		+	+	+
pTi-(PLL-PDA)_15_-(MPA-N^+^)/DEQAS		+	+	+

## Data Availability

Not applicable.

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
