# Peer review of "Influence of a Composite Polylysine-Polydopamine-Quaternary Ammonium Salt Coating on Titanium on Its Ostogenic and Antibacterial Performance"

_molecules, 2023, doi:10.3390/molecules28104120_

Round 1

Reviewer 1 Report

Please, check the attached file.

English accuracy must be improved for better readability of the manuscript. 

Reviewer 2 Report

The relevance or novelty of the study is not clarified in the document. In which it is intended to develop a material based on composite polymers (as shown in the title), but in the abstract this part is not clear. An impact or application of interest is mentioned, on the one hand paying particular attention to corrosion phenomena and apparently a relationship with affinity properties with Antibacterial/Osteogenic Properties. It is advisable to redo the summary and adapt with respect to the title, highlighting the line of study of relevance.

According to what is indicated in the state of the art, the relevance of this study seems to be oriented to the development or modification of a surface of Ti, through a technique defined and detailed by the authors. Where the results are oriented to explain the antibacterial capabilities, cytotoxicity, cell adhesion and proliferation. Therefore, it would be advisable to expand the scientific information in this regard.

Much of the results are oriented to the characterization of the composite material developed. For example, the wettability study indicates that the above results show that PLL and PDA have successfully covered the porous Ti surface and formed a hydrophilic surface. Expand on the relevance of this result, with respect to the application of interest. With appropriate supporting scientific literature.

The various results obtained are interesting, it is advisable to correlate between them and with the application of interest for the study.

Some other results could be improved by extending the analysis with adequate scientific support. As indicated in section 2.8. For example, it is indicated that the coating grafted with quaternary ammonium salt had excellent antibacterial properties compared to the metal Ti. What is the reason for this conclusion, what properties are those that allow such behavior? Expand the analysis, based on the various studies presented in the study.

Why is it that the antibacterial property of the MPA-N+ grafted coating was clearly greater than that grafted with DEQAS?. It is mentioned that it is due to anti-inflammatory advantages, and the grafting rate of two quaternary ammonium salts was relatively higher than that of one quaternary ammonium salt. Explain about it, to clarify conclusion.

Conclusions can be improved by presenting a better analysis of the results with adequate scientific support.

Author Response

请参阅附件。

Reviewer 3 Report

PLEASE SEE THE ATTACHED ZIP FILE, "REVIEW-MOLECULES-APRIL-2023.ZIP".

The quality of the English is average. As such, the manuscript would require extensive editing, which, as a service to the authors, I have done.

PLEASE SEE THE WORD FILE, "WORD-VERSION-EDITED.DOCX", which is enclosed in the ATTACHED ZIP FILE, "REVIEW-MOLECULES-APRIL-2023.ZIP".

Reviewer 4 Report

There are some points that should be clarified before considering again this manuscript 

1/ The introduction part should be improved.  The aim of the work is not well highlighted. 

2/ In the materials and Method section : The purity of all the materials used in this work should be added.

3/How do the authors explain the increase in the absorbance with time? More explanations are needed in this part.

4/The porosity of the speciemens should be provided. 

5/The authors should compare their results with previous works. 

6/ In the conclusion part the authors should add limitations and future work for this study.

Overall, English language is acceptable. Some minor grammatical and typo mistakes should be corrected. 

Round 2

Reviewer 1 Report

The revised manuscript looks better after the corrections.  Please avoid the typo and grammatical errors still present.

Open questions and aspects still to be clarified in the manuscript:

(1) A N-O bond is assigned by XPS measurements (aprox. 405 eV). However, the supported references [36-38] do not assign this group in such coatings.

Why the N-O group should be present in the coating structures studied in this work?  Apparently, it shouldn´t. 

Authors must clarify this aspect in the manuscript. Otherwise, they should avoid making speculative or incorrect assignments. 

(2) Authors missed to answer the following questions (lines 266-270):

Substrate inmersion is firstly carried out in the PLL solution and subsequently in the DA solution? or is the substrate inmersed in a mixed solution of both PLL and DA?

What is the inmersion time in the PLL solution?

What is the inmersion time in the DA solution?

or what is the inmersion time in the mixed PLL and DA solution?

What is the method used to transform DA into PDA for the LBL assembly? UV-irradation or under dark conditions? What time was finally chosen for  the formation of the PDA polymers used in the PLL-PDA assembly (Section 3.3)?

This should be properly stated in the manuscript to replicate the methodology and help reproducing the results.

(3) Section 3.8

Authors claimed that the reviewer´s comments were revised, but it seems they forgot to include them.

Please, include the following information:

-The substrate used in the QCM experiments (e.g. Si, Ag, Au, etc. on quartz?). If titanium were not used in the QCM experiments, please justify how the results could be extrapolated to titanium samples employed in the work.

-Surface area of the substrate, nominal oscillation frequency of the quartz crystal, the calibration constant for the calculations and equations used to transform frequency changes into mass.

Minor editing of English language is required

Author Response

Dear Sir or Madam:

Attached please find our revised manuscript (molecules-2378354 R1) entitled “Influence of a Composite Polylysine-Polydopamine-Quaternary Ammonium salt Coating on Titanium on its Ostogenic and An-tibacterial Performance” submitted to the journal molecules.

Thank you very much for your kind reading of our manuscript and helpful comments. Now, the manuscript is revised under the guidance of the comments. The details of how we revised our manuscript and the response to the comments are given as follows:

(1) Comment

A N-O bond is assigned by XPS measurements (aprox.405 eV). However the supported references[36-38] do not assign this group in such coatings. Why the N-O group should be present in the coating structures studied in this work? Apparently, it shouldn't. Authors must clarify this aspect in the manuscript. Otherwise, they should avoid making speculative or incorrect assignments.

Response According to your opinions and relevant literature, there is indeed a speculative error in the processing of N-O peaks. After careful consideration, Figure 5 (b), (c) and related discussion have been deleted. During this period, I have a deep understanding of XPS spectrum analysis. Hopefully with the correction made it is more clear and understandable for it’s readers.

(2) Comment

Authors missed to answer the following questions (lines266-270): Substrate inmersion is firstly carried out in the PLL solution and subsequently in the DA solution? or is the substrate inmersed in a mixed solution of both PLL and DA? What is the inmersion time in the PLL solution? What is the inmersion time in the DA solution? or what is the inmersion time in the mixed PLL and DA solution? What is the method used to transform DA into PDA for the LBL assembly? UV-irradation or under dark conditions? What time was finally chosen for the formation of the PDA polymers used in the PLL-PDA assembly (Section3.3)? This should be properly stated in the manuscript to replicate the methodology and help reproducing the results.

Response According to your comment, the specific experimental steps have been added in Section 3.3.

(3) Comment

Section 3.8: Authors claimed that the reviewer's comments were revised, but it seems they forgot to include them. Please, include the following information: The substrate used in the QCM experiments (e.g. Si, Ag, Au, etc. on quartz?). If titanium were not used in the QCM experiments, please justify how the results could be extrapolated to titanium samples employed in the work. Surface area of the substrate, nominal oscillation frequency of the quartz crystal, the calibration constant for the calculations and equations used to transform frequency changes into mass.

Response According to your comment, I am very sorry for the omission before, now add the relevant content in Section 3.8.

Thank you very much for your good job again. Any question/information please corresponds to the CORRESPONDING AUTHOR Jing Xu.

                                        Sincerely

                                        Jing Xu

                                        E-mail address: xujing@sdili.edu.cn

Reviewer 2 Report

The changes made to the document have improved the technical quality of the document. It is recommended to publish in the current form, consider improving the quality of the figures (texts or little blurred image).

Author Response

Dear Sir or Madam:

Attached please find our revised manuscript (molecules-2378354 R1) entitled “Influence of a Composite Polylysine-Polydopamine-Quaternary Ammonium salt Coating on Titanium on its Ostogenic and An-tibacterial Performance” submitted to the journal molecules.

Thank you very much for your kind reading of our manuscript and helpful comments. Now, the manuscript is revised under the guidance of the comments. The details of how we revised our manuscript and the response to the comments are given as follows:

(1) Comment

The changes made to the document have improved the technical quality of the document. It is recommended to publish in the current form, consider improving the quality of the figures (texts or little blurred image).

Response According to your comment, English writing and quality of the figures  were improved to the best of our ability in the revised manuscript. Hopefully with the correction made it is more clear and understandable for it’s readers.

Thank you very much for your good job again. Any question/information please corresponds to the CORRESPONDING AUTHOR Jing Xu.

                                        Sincerely

                                        Jing Xu

                                        E-mail address: xujing@sdili.edu.cn

Reviewer 4 Report

The authors replied reasonably to the raised comments during the first revision. However, there still some minor rectifications and typo mistakes that should carefully revised and corrected:

As an example, In the abstract: Please check this sentence" Ti specimens were surface was negatively ionized using a hot alkali activation 16 method", After careful scrutiny of the entire manuscript , the paper can be accepted for publication.

Some minor typo mistakes should be corrected before publication.

Author Response

Dear Sir or Madam:

Attached please find our revised manuscript (molecules-2378354 R1) entitled “Influence of a Composite Polylysine-Polydopamine-Quaternary Ammonium salt Coating on Titanium on its Ostogenic and An-tibacterial Performance” submitted to the journal molecules.

Thank you very much for your kind reading of our manuscript and helpful comments. Now, the manuscript is revised under the guidance of the comments. The details of how we revised our manuscript and the response to the comments are given as follows:

(1) Comment

The authors replied reasonably to the raised comments during the first revision. However, there still some minor rectifications and typo mistakes that should carefully revised and corrected: As an example, In the abstract: Please check this sentence"Ti specimens were surface was negatively ionized using a hot alkali activation 16 method, After careful scrutiny of the entire manuscript, the paper can be accepted for publication.

Response According to your comment, “Ti specimens were surface was negatively ionized using a hot alkali activation method” was revised.

Thank you very much for your good job again. Any question/information please corresponds to the CORRESPONDING AUTHOR Jing Xu.

                                        Sincerely

                                        Jing Xu

                                        E-mail address: xujing@sdili.edu.cn